# Non-Viral Delivery of Gene Therapy to the Tendon

**DOI:** 10.3390/polym14163338

**Published:** 2022-08-16

**Authors:** Jing Jin, Qian Qian Yang, You Lang Zhou

**Affiliations:** Hand Surgery Research Center, Research Central of Clinical Medicine, Affiliated Hospital of Nantong University, Medical School of Nantong University, Nantong 226001, China

**Keywords:** tendon, gene therapy, nanotechnology, non-viral vectors, nanoparticles, hydrogel

## Abstract

The tendon, as a compact connective tissue, is difficult to treat after an acute laceration or chronic degeneration. Gene-based therapy is a highly efficient strategy for diverse diseases which has been increasingly applied in tendons in recent years. As technology improves by leaps and bounds, a wide variety of non-viral vectors have been manufactured that attempt to have high biosecurity and transfection efficiency, considered to be a promising treatment modality. In this review, we examine the unwanted biological barriers, the categories of applicable genes, and the introduction and comparison of non-viral vectors. We focus on lipid-based nanoparticles and polymer-based nanoparticles, differentiating between them based on their combination with diverse chemical modifications and scaffolds.

## 1. Introduction to the Use of Non-Viral Vectors in Tendon Injuries

Tendons are very particular connective tissues with a hierarchical structure that is mainly composed of collagen and are parts of the musculoskeletal system that transmits forces from muscle to skeletal [1,2].

There are two main tendon disorders: chronic and acute injuries. Chronic injuries, also called tendinopathies, are mostly caused by the overuse of tendons or are side effects of aging. Acute injuries always occur after physical damage, such as vigorous sports activities, physical overloading, and occupational injuries [3]. A large variety of clinical statistics suggest that direct lacerations are probably more likely with certain tendons, such as the extensor and flexor tendons of the finger and hand, which are significant to their daily activities [4]. Unfortunately, hypovascularity, hypocellular, and low quantity of growth factors in tendon tissue contribute to its insufficient healing capacity [5]. Surgical repairs that aim to connect the two sides of a ruptured tendon are the most practical and common clinical therapy for treating tendon rupture [6]. However, the effects are often not as good as expected due to the tendon’s low healing potential and a series of complications, such as adhesion, scarring, and failed repairs [7].

For these reasons, the proper healing of injured tendons is the subject of a considerable number of studies. Following research and discoveries made in the last century, various methods to promote tendon healing have been proposed, such as the use of nonsteroidal anti-inflammatory drugs, cell therapy, tissue engineering, gene therapy, platelet-rich therapy, etc. [8]. Among these, gene-based therapy, a more accurate strategy, is considered one intentional method with the most efficient modulation of biological expression [9]. In 1990, the United States approved clinical trials in gene therapy for the first time. Five years later, in 1995, adeno-associated viruses (AAV) were first used, and lentiviruses were produced in 1996 to deliver genes [10,11]. Accompanied by the development of CRISPR/Cas9 gene-editing technology, gene-based therapy has come into more regular use [12]. An increasing number of trials and experiments have been conducted in clinical settings, and multiple subjects have been approved. Gene therapy, known as the one-step healing method, is applied in diverse diseases which are not limited to monogenic diseases but extend to some life-threatening conditions, such as hematemesis, ophthalmopathy, inherited metabolic disorders, neuromyopathy, etc. [13]. However, cases of severe toxicity and death remained, which prevented the Food and Drug Administration (FDA) from giving its backing to the growing clinical application of gene-based therapies [14].

In 2022, gene therapy rejuvenates using a method based on the innate characteristics of diseases [15,16]. Notably, Zolgensma has been approved in clinical trials [17]. In addition, AAV-RPE65 has also been accepted [18]. This remedy introduces exogenous nucleic acids, such as DNA, mRNA, siRNA, miRNA, or ASO to the targeted position in vivo or in vitro. The main mechanism is substituting pathogenic genes with normal ones or silencing them [19].

The rationale behind the use of gene-based therapy in tendon injuries is that the critical genes that promote tendon healing will not be produced immediately after injury. What is more important is that genes for the formation of adhesion will be overexpressed. Tendon injuries invariably result in an ill-balanced process. The aim of using gene-based therapy is an attempt to remedy this disequilibrium and gain the highest healing with the rare formation of tendon adhesion after repair [6].

Two methods may be used to deliver genetic materials into sites of interest: (1) in vivo gene-based therapy, which is directly combined with the injured sites and releases the therapeutic factors, and (2) ex vivo gene-based therapy, which implants cells into defective tissues after genetic editing [20]. Due to the hypocellularity of the tendon tissue, we focus on in vivo gene-based therapy, which is a simpler and cell-free application [21].

At present, there are two kinds of vectors used in gene therapy: viral vectors and non-viral vectors. Viral vectors that include adenovirus, adeno-associated virus, retrovirus, and lentiviruses, possessing high transfection efficacy and stable gene expression, are major research topics [22]. However, there are still several disadvantages, like immunogenicity, insertional mutagenesis, oncogenesis, fatal feasibility, lack of appropriate targeting, limited capacity for packaging genes, and the difficulty of production in the viral vectors limiting medical translation [23,24,25,26,27,28,29]. Despite these disadvantages, viral vectors remain the most frequently used in delivery systems [30]. Gene therapy based on non-viral vectors provides better biological safety. Meanwhile, it is an ideal method to carry longer genetic segments and is more easily deployed [31]. However, compared to viral vectors, the disadvantages such as impediments in breaking through the physical barriers and low transfection efficiency remain [20]. With the development of material science, there has been a growing number of publications focusing on the application of novel materials in gene therapy. As shown in *Nature Biotechnology*, the non-viral vectors based on research show a promising tendency, and half of the relevant patents have been approved.

Based on our scrutiny of gene therapy, and the gene and non-viral vectors, in particular, the transfection efficiency and the volume of injection remain to be crucial for healing properties [6]. In this review, we will sum up the barriers to delivering nucleic acids to the tendon in vivo and focus on how novel non-viral vectors overcome these obstacles. Finally, we will discuss whether these non-viral vectors can be transformed clinically.

## 2. Biological Barriers

No matter what kinds of genes are utilized in gene therapy, they will always be detected by the immune system as Figure 1 shows, as a result of the chemical modifications of carried genes and the intrinsic encapsulation of vectors. Besides, the extracellular endonucleases act as scissors to speed up the process of degradation. The cell membrane serves as a physical barrier that usually mixes with drugs via endocytosis. When they enter the cytoplasm as endosomes, vectors and carried genes are easily degraded by the clearance of lysosomes. Finally, a few genes are likely able to enter the nucleus and play the role of overexpression or silencing of target genes.

Tendons are compact structures mainly composed of hierarchically structured collagen, with low cellularity and hypovascularity [1]. Their instinctive characteristics lead to the insufficient effects of gene therapy. The capacity of the injected agents is limited by the special composition, and there is also a high possibility of leakage [32]. Microneedle, as a conventional method of injection, was employed in a broad variety of research; however, it was thought to have trouble controlling the action of related injuries [33]. Above all, a vehicle for gene delivery possessing the ability to release genes sustainably and store ample drugs is critical to improving the curative effect.

Overall, effective vectors should possess the following merits:Avoid the degradation of endonucleases and the detection of immune systems;Help genes enter cells through endocytosis mediated by receptors;Promote lysosome escape;Release at a sustainable speed and be able to entrap enough drugs;The basic quality of biocapacity, biodegradation, and non-toxicity.

## 3. Different Forms of Loaded Genes

### 3.1. DNA

Previous studies have shown that extraneous DNA will be sensed by Toll-like receptor 9 (TLR9), a pattern recognition receptor [34]. Many clinical trials have been performed using DNA-based gene therapy. However, endonucleases in the extracellular matrix are barriers to DNA delivery. Concerning the characteristics of rapid degradation in vivo and resistance to the negative charge, naked nucleic acids have trouble passing through the cell membrane [35].

Last but not least, it is extraordinarily vital for DNA to enter the nucleus through its outer membrane. Previous studies have reported that no corresponding activity could be detected with the direct microinjection of DNA into the cytoplasm. Conversely, when injected into nuclei directly, the expressions of targeted genes will represent 50–100% of the nuclear area [36].

### 3.2. RNA

RNA-based gene therapy can restrain the side effects of insertional mutagenesis which would occur in DNA-based gene therapy. However, RNA molecules are anions and highly sensitive to intracellular RNases. The establishment of delivery systems is vital to improving the therapeutic effects of RNA drugs. However, owing to cytotoxicity and undesirable efficiency, many carriers are still under study, and steps toward clinical transformation have been postponed [37]. As the COVID-19 pandemic is raging, LNP-based gene therapy appeared and exhibited a commendable anti-infection function in the lung, which simultaneously roused the masses from their nearly lost interest [38,39].

### 3.3. mRNA

Unlike DNA, gene therapy based on mRNA avoids the necessity of passing through the nuclear membrane and elicits the secondary action of gene integration [30,40]. Thus, it has been extensively investigated in clinical studies extensively [41,42,43,44].

Nevertheless, there still exist barriers to mRNA delivery. This is firstly due to the relatively large molecules of mRNAs being able to penetrate through the cell membrane on their own, being attracted to water and negatively charged, making a considerable obstacle to overcoming physical carriers. Secondly, mRNAs are susceptible to RNases present in tendon tissues and the facile interaction between the cationic and anionic charge makes it tough for large molecule mRNAs to enter cells and kick in [37].

The mRNAs with no modifications can trigger the interaction with Toll-like receptors (TLRs), and lead to serious toxicity [45,46]. Chemical modifications with 2-thiouridine and 5-methylcytidine could alleviate the immune response via TLR3, TLR7, TLR8, and retinoic acid-inducible gene I (RIG-I) [47]. Overall, chemical modification and the combination of nanoparticles are still the preferred methods for overcoming this impediment.

### 3.4. siRNA

RNA interference (RNAi) is a promising technique and it has superior potential in gene therapy [48]. The siRNA can silence almost any target gene by reducing detrimental proteins before synthesizing, which interacts with mRNA directly. The peculiar mechanism in siRNA makes it an ideal method to be applied to numerous diseases [49].

The siRNAs are commonly 19–21 bp in length and they are typically synthesized double-stranded RNAs [50]. However, during in vivo delivery, unmodified siRNAs will be hindered by endonucleases in tissues and initiate immune action [51,52,53]. A wide range of siRNA-based studies in tendon injury models have demonstrated that it is scientific to use siRNA as a drug in the regulation of tendon disorders [54]. We list the siRNAs that have been used in tendon healing models in Table 1.

Although siRNAs are delicate, being modified with chemical moieties could provide them with a stable structure to decrease the possibility of degradation and the occurrence of side effects [55]. In addition, entrapping siRNAs in porous nanoparticles also provides a shield from unwanted degradation and immune recognition [56].

### 3.5. miRNA

miRNAs are small endogenous non-coding RNAs around 22 nucleotides that regulate gene expression, including inflammation, cell cycle regulation, cell proliferation, apoptosis, death, etc. [88,89,90,91]. There has been growing interest in miRNA-based drugs since a broad range of evidence has demonstrated that miRNAs work as fundamental regulators in many physiological mechanisms [92]. In addition, several studies have found that protein-coding genes regulated by miRNA were more than 30% of the total, and aberrant miRNAs were expressed in many pathema [93]. In recent research, miRNAs have also shown participation in tendon repair. The functions of miRNAs in tendon disorders can be classified into the various biological system processes:

Cell viability: CUGBP2 and MYB, which participate in the regulation of cell apoptosis, cell proliferation, and apoptosis, were considered to be regulated by miRNA-499 [94]. The miR-205-5p was assured to enhance the expression of VEGFA genes, which promotes tendon healing by improving cell viability [95].

Inflammation: Inflammation has been considered throughout the whole process of tendon healing. The JAK2/STAT3 pathway, AMPK, and TREM-1 signal pathway were reported to be related to inflammation. The first pathway can be inhibited by miR-146a-5p, while the latter can be inhibited by miR-31-5p, miR-195-5p, etc. [96,97].

Adhesion formation: As a troublesome complication after tendon repair, adhesion is so remarkable that a wide range of explorations have been performed. The miR-29b was deemed to inhibit the growth of fibroblasts via TGF-β and Smad3 [98].

Owing to the great similarities between microRNAs and siRNAs in the terms of structure, charge, and molecular weight, the delivery barrier of microRNAs would be extraordinarily similar to that of siRNAs. The synthesized delivery systems for siRNAs could be applied in the transfection of miRNAs [50].

## 4. Non-Viral Vectors for Gene Therapy and Feasibility Analysis

Although all of these genes can be more stable with chemical modifications, various non-viral delivery systems have been constructed to alleviate the consumption in the process of transfection and we have concluded them in Figure 2. Non-viral vectors are not only less expensive than viral vectors, but also more convenient to construct.

As mentioned above, there are several qualities that vectors should possess in terms of transfection efficiency and biological characteristics. We will list several kinds of vectors that have been widely synthesized in recent years and focus on how novel non-viral vectors overcome these obstacles, which are classified into delivery barriers. In addition, we will also focus on the carrying ability and speed of releasing drugs. Based on this, we will conduct a feasibility analysis. For a clear representation, we have drawn a graphical abstract figure of this article.

## 5. The Plasmid

The plasmid, as a circular double-stranded DNA, could replicate which is independent of the chromosome and can transfect genes of interest into cells [99]. The plasmid is a milestone in the non-viral vectors for gene delivery. In the 1990s, the first exploration of plasmids had the potential to be used as non-viral vectors [100]. The expression of the transgene and the production of proteins of interest are required for function. Plasmids are routinely used as expression vectors in non-viral gene therapy studies owing to their ease of construction and amplification. Moreover, plasmids are episomal and non-integrating, which reduces the risk of insertional mutagenesis compared with viral vectors. Last but not least, plasmids can be used repeatedly, making them cost-effective [101]. Due to the traits mentioned above, plasmids have entered clinical trials.

Even so, plasmids contain some inherent limitations. Plasmids are less sufficient than viral vectors to deliver their payloads, not to mention vectors like polymer-based nanoparticles and lipid-based nanoparticles [102,103]. In addition, the likelihood of genome mutation is increased. A bacterial origin of replication, existing in plasmids, which propagate in host cells, will generate potential side effects similar to the mechanism of antibiotics [101]. For better clinical transformation, the plasmid should be modified by deleting unwanted sequences [104]. Studies on tendon injuries usually use plasmids to carry genes and encapsulate them into novel delivery tools which we will introduce later.

## 6. Exosomes

Exosomes are nanosized vesicles around 40–100 nm released from various types of cells, functioning as material carriers and signal communicators [105]. Having a similar structure to cell membranes that constitute endogenetic lipids, proteins and ribonucleic acid makes it fuse completely with target cells. They are not only natural carriers of RNA, but also abundant in body fluid, which makes them convenient to obtain [106]. Notably, a study on gene transfer into the murine retina showed that exo-AAV-GFP vectors were more efficient than conventional AAV-GFP, providing an exosome-based gene transfer method with great prospects [107]. However, the biological functions of exosomes from different cell types vary greatly, which means that the possibility of their application also varies. Meanwhile, the likelihood of tumorigenesis and immunosuppressing also exist, making the reliability of transformation questionable.

## 7. Inorganic Nanoparticles

Certain advantages, such as convenient synthetic methods, large material storage, stability, and ease of modification, make inorganic nanocarriers promising for application in gene therapy. However, the question of whether inorganic nanocarriers cause threats to organisms still needs in-depth discussion. At present, gold nanoparticles (AuNPs), Silver nanoparticles (AgNPs), carbon nanotubes (CNTs), mesoporous silica nanoparticles (MSN), graphene, and up-conversion nanoparticles (UCNs), have already been designed and put into widespread use [108,109,110]. We will demonstrate the application of inorganic nanoparticles below and discuss in greater depth cell toxicity which limits their utilization.

### 7.1. Gold

AuNPs are classically used nanocarriers, possessing good stability and high gene loading, that can be controllably modified and effectively applied in gene therapy [111]. However, it is tough for gold nanoparticles to have advanced transformations in clinical trials. Due to the different establishments of AuNPs based on their diverse shapes, sizes, etc., there has been a division among researchers over whether AuNPs demonstrate toxicity. Beyond that, recent research has demonstrated that AuNPs would promote cell apoptosis via reactive oxygen species (ROS) interactions. For example, Elizabeth R conjugated carboxylic-terminated gold nanoparticles (AuNP) and polydopamine (PAMAM) dendrimers by EDC and sulfo-NHS, as delivery systems. The sMUA–AuPAMAM complexes demonstrated hypothetical transfection efficiency, stability, and enough DNA encapsulation, while the toxicity remained [112].

### 7.2. Ag

Silver (Ag), well known for its anti-bacterial and anti-inflammatory properties, was expected to be a suitable tool for tendon tissue engineering [113]. However, AgNPs may also lead to poor cell viability and proliferation [114,115,116]. Many aspects of the formation of tendon adhesion are due to infection caused by bacteria. Despite the occurrence of toxicity, some consider that AgNPs may still deserve further research on the natural character of their anti-bacterial function. To determine whether AgNPs have any cytotoxicity, a team worked on them and found that AgNPs could lead to some processes related to Tendon-derived stem cells (TDSCs) apoptosis, such as the generation of active oxygen and the depolarization of the mitochondrial membrane. The results were not favorable for the use of AgNPs to promote tendon healing [117].

### 7.3. Silica

Mesoporous silica nanoparticles (MSN), have a porous makeup consisting of hundreds of mesopores, which can store nucleic acids and release them at a sustainable speed [118,119]. Since the first production of mesoporous silica nanoparticles (MSN) in 2004, cationlization with PAMAM dendrimers has shown considerable influence on HeLa cells, and there is a steady stream of syntheses of complexes carrying nucleic acids [120,121]. The majority of MSNs have been modified with polymers, such as poly-L-arginine or polyethyleneimine (PEI) [122]. PEI is positively charged with internal PH and it has a strong potential to interact with nucleic acids. Nevertheless, PEI/poly-L-Lysine-modified MSNs have been described as having cytotoxicity [123,124]. Nevertheless, imidazole and amino groups have both been used in the functionalization of MSN.

Arnaud Suwalski and colleagues delivered the PDGF-B gene via amino- and carboxyl-modified mesoporous silica nanoparticles into rat Achilles tendons. While in vitro experiments designed for primary tenocytes assured deficiency in transfection, all biomechanical tests and tendon adhesion scores in vivo demonstrated good tendon healing. Remarkably, the inflammation interaction and necrosis were almost undetectable. The activity of the luciferase reporter gene-encoding plasmid was sustained for at least 2 weeks [123]. To compare the therapeutic effects of NH_2_- and His-, David Brevet designed histidine-functionalized mesoporous silica nanoparticles in Achilles tendons in cells and in vivo. The results showed that MSN-His possesses a better ability to avoid DNase degradation and has higher transfection efficiency than MSN-NH_2_. In addition, and more importantly, MSN-His avoided the limitation of toxic effect in pDNA/MSN-NH_2_ [124].

## 8. Lipids and Lipid-Based Nanoparticles

Lipids have been designed for gene therapy, sharing similar structures of major components in cell membranes, and are assumed to have the capability to carry materials such as genes and proteins into various cells [125]. The compacted complexes formed by the electrostatic interaction between lipids and target genes were called lipoplexes. As Figure 3 shows, these nanoparticles have three well-defined structures: (1) micelle; (2) liposome; (3) lipid-based nanoparticle. The micelle-monolayer and liposome-bilayer are both classified as Lipid-based structures [126].

Produced by Phospholipid biomolecules, liposomes can entrap both lipid-soluble and water-soluble molecules well and deliver them to the target sites by mixing them with the cell membrane [20]. They all consist of three main domains: a hydrophilic cationic (ionizable) headgroup, a hydrophobic tail group, and a linker group. In 1980, the first use of liposomes composed of phosphatidylserine for DNA delivery triggered a large variety of lipid-based research [127]. In 1987, the term Lipofection was first described by Felgner. A kind of cationic lipid, N-[1-(2,3-dioleyloxy) propyl]-N, N, N-trimethylammonium chloride (DOTMA) was synthesized, which showed 100% entrapment of the DNA. Compared to calcium phosphate and diethylaminoethyl dextran (DEAE dextran), it proved to have almost 100-fold more transfection efficiency and worked stable in vivo. Finally, the DOTMA-DNA complexes with a concentration of 50–100 ug that depend on cell type show no significant cytotoxicity [128]. After decades of discovery, three categories have been identified: amino lipids, optimized ionizable lipids, and lipidoids. To improve the workpiece ratio, a broad variety of non-viral vectors based on liposomes have been produced, including exosome–lipid nanoparticles, hyaluronic acid (HA) -modified cationic lipids, etc. [129].

Lipid-based nanoparticles (LNPs) are more complex, with multiple lipid layers, showing the merits of biocompatibility, less cytotoxicity, better drug affordability, and decreased endosomal escape [130]. At present, Lipofectamine™, TurboFect™, and Stemfect™ have been put on the market. Among them, one is for siRNA delivery, and the other two are mRNA-based vaccines for the treatment of COVID-19 [37]. As the following picture 2b shows, Lipid-based nanoparticles (LNPs) contain four fundamental parts: (a) a cationic lipid or an ionizable lipid; (b) cholesterol; (c) a helper lipid; (d) a poly (ethylene glycol) (PEG)-lipid.

In addition, the composition is more effective than lipid-based structures and decreases the dose needed to achieve purpose therapeutics. Abundant researchers have found that all these components have great effects on delivery. The activity of the LNP-based gene delivery systems is strongly associated with the value of PH, which influences stability and encapsulation capacity [131]. In an acid microenvironment, LNPs combine with negative RNAs, which act as positive constituents, showing a higher encapsulation efficiency. However, the positive charge also harms cells, which is largely overcome by the synthesis of ionizable lipids. Ionizable lipids that are neutral in mild cytoplasm will be protonated in acidic endosomes, which at the same time promote the release of target genes and increase endosomal escape. It is generally believed that the PKa of Ionizable lipids varies from 6.2 to 6.5, ensuring sufficient encapsulation efficiency and stability [132]. Meanwhile, the PEG-lipids are anchored in LNPs with the help of lipids, and the barriers of water are constructed via the hydrophilic interaction of PEG, which is like PEG-modified nanoparticles. The structure consists of a bilayer and PEG, cholesterol, or DOPE, which is known as a stable nucleic-acid lipid particle (SNALP). PEG-lipid, as a shell outside, could protect genes inside the core of nanoparticles and increase biological safety. Moreover, PEG is hydrophilic and forms a protective layer on the surface of nanoparticles, preventing complexes from recognition by immune systems. Finally, SNALPs show a sustainable release [133].

Collagen, as the dominant constituent of the tendon tissue, is predicted to promote tendon healing with the direct overexpression produced by gene therapy. PDGF-B cDNA directly injected into vivo also promotes collagen production and enhances angiogenesis, showing a promising therapeutic gene for tendon repair [56]. Overall, it is certain that PDGF-B would promote the synthesis of collagen, which may enhance the strength of the injured tendon. Based on these predicted theories, Wang and colleagues established a complex with 15 ug plasmid containing the PDGF-B complementary deoxyribonucleic acid (cDNA) and 60 uL Lipofectin (Invitrogen, Carlsbad, CA, USA) and transfected into rat intrasynovial tenocytes, which expressed 25% type I collagen genes more than the control group, and notably improved tendon strength [134]. The results deduced in this research confirm the predicted theories and affirm that lipofection would be a promising tool for enhancing tendon healing. In addition, Liu, transfected miR-378a mimics with Lipofectamine 3000 (Thermo Fisher Scientific, Waltham, MA, USA), which indicated that it indeed plays a vital role in suppressing the production of collagen and extracellular matrix (ECM) both in vivo and in vitro [135]. The examples listed above all speak in favor of the application of LNPs in gene therapy, having sufficient transfection and encapsulation efficiency.

However, it is still uncertain whether the lipids-based delivery systems are superior to viral vectors or other non-viral vectors. To verify the superiority of lipids-based delivery systems, ample research was conducted. For example, J. Park and colleagues distinguish between the liposomes and adenoviral vectors which are currently occupying a large market in gene therapy, in BMP-2 gene transfer. The results showed a similar transfection efficiency, while the liposomes have the advantage of being easier to produce and able to load longer gene sequences [35]. In addition, Anthony Delalande and colleagues compared the effects of histidylated liposomes (Lip100) and histidylated linear polyethylenimine (PTG1) in gene therapy. They construct vectors with the fibromodulin (FMOD) gene, PDGF gene, and Lum gene to transfect the tendon healing model both in vivo and in vitro. The results displayed that Lip100 showed more than 100-fold transfection efficacy in vivo, while the PTG1 possessed 30% efficacy in vitro higher than Lip100. Meanwhile, the Lip100 complex with the FMOD gene, which is a significant participant in proteoglycan, with a ratio of 3:1, played a vital role in establishing a well-organized matrix structure and enhancing tendon stiffness [136].

Although the advantages of LNPs make them prominent in the comparison with other delivery systems, the successful application of mRNA-LNP-based vaccine which is synthesized to combat COVID-19 shows a promising future. However, the inherent physiological characteristics of tendon tissue are so extraordinary that the unsophisticated LNPs used in isolation could not conform to their demands. The dense structure of the tendon limits the volume of liquid injection. Furthermore, the phase of tendon healing is a long period, and it is vital to utilize a sustainable method. Based on these limitations, LNPs are restricted in individual applications to injured tendons.

## 9. Polymers, Polymer-Based Nanoparticles, and Polymer-Modified Novel Tools

### 9.1. Polymers

Cationic polymers are irreplaceable parts of non-viral vectors, owing to their potential for functionalization [137]. They are constituted by highly controllable repeating construction units, which can be replaced by other short chains depending on the requirements [138]. Poly (l-lysine) (PLL) and polyethylenimine (PEI) are the first generations of the polymer class. PLL, as a consubstantial component of lysine, showed the possibility of condensing deoxyribonucleic acids (DNAs) into target tissue in the 1980s [139,140]. Since then, it has been used for gene delivery. While poor transfection efficacy and untoward degradation have been reported, a wide range of modifications intended to increase delivery properties have been applied to PLL [141]. PLL can easily combine with negative proteins in a high serum microenvironment, which would inhibit the objective integration of genes. PEGylated PLL was produced to provide treatment for bladder fibrosis and was supported by clinical trials [142,143]. With the development of chemical synthesis technology, PEI has been synthesized, possessing a nitrogen atom at every point of intersection, and is assumed to overcome the barriers of endosomal escape [144]. As mentioned in a series of previous studies, the molecular weight and structure, linear or branched, all contribute to the advantages of PEI [37,145,146]. For example, the highly branched polymers, poly (β-amino esters) (BPAE-NB), revealed better transfection efficacy of DNA/siRNA than PEI 25k and Lipofectamine 2000, which have been available commercially already [147]. In addition, PEI is used most widely as a condenser, which usually plays a role in chemical modification [148]. However, toxicity was also found in PEI and promoted the combination of PEG and PEI [112,149]. It has been acknowledged that the most suitable N/P ratio was 10, which represents the nitrogen in PEI to the phosphorus in genes. When the ratio was beyond 10, the PEI displayed extra toxicity. Conversely, the established delivery system will not encapsulate enough target drugs [134]. Chitosan, which consists of polysaccharides, is also highly compatible and can be modified with -NH3 and -OH for better efficiency. It is not easily dissolved in water, although this will be improved by the addition of PEG and hyaluronan [19].

Although some insufficient aspects were displayed in the polymers mentioned above, the shortcomings can be overcome by controllable modifications [139]. Poly [ (2-dimethylamino) ethyl methacrylate] (pDMAEMA), poly (β-amino ester) s (BPAE-NB), and various polymers based on carbon were evaluated in clinical trials [144]. Polyamidoamine (PAMAM), a high-branching cationic synthetic dendrimer, with a positively charged amino radical on the surface, can connect the negatively charged substance on the surface of cells. Gu et al. constructed GO-PAMAM dendrimers to deliver doxorubicin (DOX) and MMP-9 shRNA plasmids, showing notable transfection efficacy and ample therapeutic effects in turning the expression of MMP-9 proteins down in MCF-7 cells in breast cancer [150]. A novel cation polyphosphoramide (PPA) modified with amine was synthesized and we observed that the ratio of PPA/pDNA (3:1) showed the highest transfection efficiency. With an optimistic result obtained by the cytotoxicity assay, the GFP-PKD2 gene transfected with PPA could also activate the NF-kB signal pathway [76,151].

### 9.2. Polymer-Based Nanoparticles

Hyun-Ji Park established amine end-modified PBAE nanoparticles, which show higher transfection efficacy than Lipofectamine 2000, carrying Sonic hedgehog (SHH) gene in vitro gene therapy, and found that it may greatly promote wound healing [152].

The TGF-β1, an isoform of TGF-β, is widely considered to be associated with adhesion formation and fibrogenesis after tendon injuries, eliciting a mass of studies based on it [153]. The TGF-β1 could also upregulate the PAI-1, a major inhibitor of matrix metalloproteinases (MMPs). Margaret studied the PAI-1 in knockout mice and found that tissues around the injured site had no obvious fibrotic adhesion, meanwhile, the repaired zone II flexor tendon showed better biomechanical outcomes. When they used siRNA, which downregulates the expression of serpine1 engineered nanoparticles, the results displayed the reverse consequences. The delivery system was synthesized with diblock copolymers which consisted of a cationic block and a PH-sensitive endosmotic block and nanoparticles. Thereafter, the cationic block was synthesized with poly (dimethylaminoethylmethacrylate) (pDMAEMA), interacting with anionic siRNA. Additionally, the second group is a ternary polymer including DMAEMA, 2-propylacrylic acid (PAA), and butyl methacrylate (BMA), possessing the properties of endosomal escape. This report described NP-mediated transfection of siRNA to promote tendon healing in vitro for the first time [154]. We have designed and synthesized a double-stranded TGF-β1-miRNA which was deemed to silence the expression of TGF-β1 and loaded it onto pcDNA6.2-GW/EGFP-miR plasmids. Finally, we entrapped the plasmid complexes into the PEI-modified polylactic-co-glycolic acid (PLGA) nanoparticles. The results of mechanical tests and biochemistry all affirm the expected results [32,155].

This kind of nanoparticle-based electrostatic incorporation constructs a multilayer structure that effectively protects genes against the clearance of endonucleases and avoids toxicity accompanied by the unreasonable use of cationic polymers. Beyond that, a negative charge outside can prevent unspecific interactions with proteins. At present, another layer-by-layer structure based on poly-L-arginine, PLGA nanoparticles, and HA is a promising tool. Poly-L-arginine can load enough siRNAs and have a low ratio of N/P that form a stable nanoparticle that can avert aggregations [156].

### 9.3. Polymer-Modified Novel Tools

Polydopamine (PDA) is a mussel-inspired protein, with strong adhesive ability, which has been applied as a coating for diverse nanoparticles in recent years [157]. The PDA shell on NPs shows high stability and low cytotoxicity in vivo [158]. Poly (lactic-co-glycolic acid) (PLGA) NPs were incubated with dopamine in an alkaline environment and modified with SH- or NH2- terminated functional ligands, which finally turned into nano-drug carriers [87]. We previously constructed a kind of suture modified with polydopamine and found that it could adhere firmly to the tendon injury model even though the tissue was sutured, and the complexes showed a sustainable release. Our group used this suture to deliver pEGFP-bFGF and pEGFP-VEGFA into tendon tissues. The injected volume was not limited by the condensed structure of the tendon tissue itself. Conversely, plasmid/nanoparticle complexes tightly adhere to sutures with strong adhesion of polydopamine. With the sustainable degradation of nanoparticles, the genes were released 78% in 28 days [159]. A broad variety of mechanical protocols were experimented with to evaluate the effectiveness of nanoparticle-coated sutures carrying growth-factor genes. In the in vivo study, the ultimate strength and gliding excursion of repaired tendons was largely enhanced compared to the control group. Meanwhile, the adhesion score was significantly decreased [129].

## 10. Comparison between Lipid- and Polymer-Based Nanoparticles

With the development of Materials Science and Engineering, synthetic materials based on nanotechnology have been reported to play a notably significant role in gene transfer in tendon disorders. Nano-strategy has the merit of ensuring good distribution of the drug concentration around the injured site, whether in a temporal or a spatial dimension. Otherwise, tendons and the surrounding extracellular matrix are structurally nanostructured materials, which exactly correspond with the intrinsic specialty of nanomaterials [160].

Nanoparticles (NPs), with dimensions <100 nm, are colloidal structures, representing a milestone in the gene-loaded materials of tendon disorders. NPs are encapsulated by the target cell membrane more easily than traditional macromolecules, which demonstrate a higher transfection efficiency [161]. Compared with conventional systems, they also demonstrate a more prolonged half-life, fewer side effects, and increased therapeutic effects which act as a partial drug depot [155]. Furthermore, NPs must prolong the release of the work period of carried drugs by protecting drugs from endosomal degradation, which is essential for the prevention of tendon adhesion formation [162]. Last but not least, they can also combine with other structures and make the best of both. Nanoparticles, owing to their nano-sized structures, are endocrine by a cell membrane or through membrane translocation directly [163,164]. Many researchers have used nanoparticles in several other ways: for their anti-bacterial, anti-adhesion, and anti-inflammatory properties. In tendon injuries, several nanoparticles have been developed to be used for gene-based delivery systems [165]. As mentioned above, nanoparticles mainly include three categories: inorganic nanoparticles, lipid-based nanoparticles, and polymer-based nanoparticles. Henceforth, inorganic nanoparticles will be excluded from the discussion.

Nanoparticles need to improve encapsulation efficiency and structural stability, which could protect genes against degradation by endonucleases. Several strategies have been explored to overcome this obstacle. Among them, electrostatic adsorption has been used most frequently. Chemical modifications which consist of cationic lipids, cationic polymeric materials, and amphoteric polymer molecules, serve as cationic parts to combine with negative genes. In this way, the gene–nanoparticle complexes could avoid the clearance of endonucleases to a large extent. However, the encapsulation efficiency is still limited, and the toxicity that exists in cationic groups is unavoidable. At the same time, biological macromolecules in the microenvironment could nonspecifically bind with cationic substances [166]. Therefore, scientists have constructed a multilayer structure based on electrostatic incorporation, such as PEI-modified PLGA nanoparticles described above. This layer-by-layer structure is a stabilized delivery system that escapes the aggregation of particles no matter itself or unspecific proteins [167,168]. For the rest, the stable nucleic-acid lipid particles as core–shell constructions are also equipped with similar advantages to PEI-modified nanoparticles [169,170].

A prolonged circulation in the microenvironment of tendon tissues can improve the curative effects. The physicochemical properties that consist of size, shape, resilience, etc., play dominant roles in sustainable release [171,172,173]. Meanwhile, the modification of PEG, which would form a protective shield against the recognition by immune systems owing to hydrophily, can also improve the circulation time. However, it will also make it difficult for nanoparticles to transmembrane, which impedes the progress of clinical applications [174,175]. According to research findings, polymers based on the proton cavernous effect and lipids based on the fusion and rupture of cell membrane all promote endosomal escape [176,177]. In conclusion, we suppose that lipid-based nanoparticles and polymer-based nanoparticles have promising prospects.

## 11. Hydrogel

To help gene-loaded nanoparticles disperse more evenly around the injured tendon, some scientists have made certain efforts to explore biological materials. Hydrogel, as a three-dimensional (3D) hydrophilic polymer network which is mainly constituted of water, is similar to the physical microenvironment [178]. It is also a mesoporous polymer network that could encapsulate drugs. Furthermore, different modified hydrogels could bear mechanical strength corresponding to the target tissues. Studies have shown that a hydrogel used in tendons or ligaments should bear a tensile strength of around 10–100 MPa, fracture toughness of around 20–30 kJ m^−2^, and a fatigue threshold of around 1000 J m^−2^ [179,180]. Finally, a hydrogel could also function as an anti-adhesion barrier to prevent the formation of tendon adhesion [181]. In conclusion, hydrogel, as a tough, strong, elastic tool, has attracted a wide range of applications in tendon disorders. We have described the application of hydrogel in vivo as shown in Figure 4.

In the history of hydrogels, a wide range of substances have been used, including natural and compositive polymers. Due to their similarity with biological composition, natural polymers are more biocompatible with the body than synthesized polymers. The natural ones compromise hyaluronic acid, collagen, fibrin, agarose, chitosan, alginate, gelatin, and cellulose [182,183,184,185,186,187,188,189,190,191]. The synthesized ones include poly (acrylic acid) (PAA), poly (ethylene glycol) (PEG) or poly (ethylene oxide) (PEO), poly (vinyl alcohol) (PVA), poly (2-hydroxyethyl methacrylate), poly (N-isopropyl acrylamide), and silicon [192,193,194]. PEG hydrogels can also conjugate with biomolecules like nucleic acids and avoid interaction with target tissues or cells.

Jan Schulze and colleagues combined PEI-modified polyplexes and corresponding lipopolyplexes and encapsulated them into microparticulate PVA hydrogels, which were characterized as Nanoparticles-in-Microparticles Delivery Systems (NiMDS). By regulating parameters such as PVA physical crosslinkers, molecular weights, etc., the established hydrogels showed a target-made, the long-period release of nanoparticles [192].

Benjamin R. Freedman also constructed a novel hydrogel named Janus Tough Adhesive (JTA). JTA which is biocompatible can adhere firmly to the diseased tendon by chitosan and represents a tough gel. This novel material is an advanced drug depot that can store considerable quantities of medicine. In addition, the group has also verified the effects of JTA on several tissues, such as the patellar, supraspinatus, and Achilles tendons. No matter what sites they function, they all promote tendon healing and suppress inflammation [195].

The separation of injured tendons and peritendinous tissues is significant for preventing adhesion. Cai et al. developed an anti-adhesion barrier to inhibit peritendinous adhesion. They constructed a kind of hydrogel with good mechanical properties, which was modified by oxidized HA-containing aldehyde groups (HA-CHO) and adipic acid dihydrazide-modified HA (HA-ADH). In addition, they also established Smad3-siRNA nanoparticles and encapsulated them into MMP-degradable GelMA microspheres. The combination of these two structures accelerated the effects of decreasing inflammation and tendon adhesion [85].

It is well-known that the inflammation response initiates the formation of adhesion after tendon injury. Among the proteins that participate in inflammation, cyclooxygenases (COX-1 and COX-2) function by synthesizing prostaglandins and play an important role in the inflammation phase. However, taking nonsteroidal anti-inflammatory drugs orally does not effectively inhibit adhesion formation with the loss of rapid renal clearance and the risk of several side effects like gastrointestinal mucosal injury, respiratory tract complications, etc. [196,197,198]. Our group also developed a sustainable release platform for cyclooxygenase-engineered miRNA plasmid, which was inserted within the polyethylenimine (PEI) -modified PLGA nanoparticles and embedded in the hyaluronic acid (HA) hydrogel. They compared the levels of COX with different organizations of miRNAs and chose COX-1-miRNA1 and COX-2-miRNA2 as the downstream study group. The biocompatible hydrogel showed a sustainable release efficiency and avoided an inevitable loss in the process of transfection, with a three-dimensional network structure [199]. Then, based on our previous study, we have also assessed the treatment of hydrogel that carries the siRNA/nanoparticles in vivo and in vitro. The biomechanical tests confirmed the therapeutic effects of the tools in vivo and in vitro, and the group of tenocytes cultured with COX siRNA/nanoparticle showed a high proliferation [92].

## 12. Other Worthy Delivery Systems

Yan and colleagues constructed a double-layer tool, containing a poly (lactic-co-glycolic) (PLGA) electrospun membrane outside, and a poly (ethylene glycol) -block-poly (L-valine) (PEG-PLV) inside. The outer membrane is loaded with IBU to prevent inflammatory factors from aggregating, and the inner hydrogel is carried with bFGF to strengthen the healing of tendon tissues. The results undoubtedly confirm the therapeutic effects of this composition [156].

Deoxyribonucleic acid (DNA), as a natural substance, is considered to be a promising synthesizer for biocompatible vectors. In 1996, a group designed the first hydrogel based on DNA by crosslinking the single-stranded DNA (ssDNA) and polyacrylamide. DNA strands, as a programmable structure, can be built into a diverse network. Additionally, when it is in a three-dimension pattern, the DNA can afford mechanical elasticity and form a stable matrix [200,201]. The DNA hydrogel was synthesized via chemical and physical crosslinking, the former refers to covalently synthesized linear DNA-DNA and DNA-polymer, and the latter corresponds to noncovalent interactions. As an interdisciplinary combination, the DNA hydrogel takes pride in its comprehensive merits, such as sufficient encapsulation efficacy, sensitive molecular recognition, etc. In recent years, gene-loaded DNA hydrogels have been widely developed and have shown high biosafety. The DNA hydrogel no doubt deserves further exploration in clinical transformation [202].

Wu and colleagues used three-dimensional printing technology to construct a tendon structure mimic. They also loaded microRNA which silences the TGF-β1 gene PDA nanoparticles and puts the complexes into the established tendon scaffold. All biomechanical tests and histology affirm the functioning of this delivery system [203].

## 13. Summary and Prospect

Since the start of the COVID-19 pandemic, vaccines based on lipid-based nanoparticles have been applied successfully, showing that a new era of gene therapy has begun. By now, a considerable number of governments have recognized the promising future of gene therapy. With the burgeoning development of nanotechnology, more and more non-viral vectors have appeared and shown a great many advantages. However, due to several obstacles such as biological barriers and the elusive balance between transfection efficiency and toxicity, vectors that have been successfully marketed are few. Tendons, as dense tissues, require high payloads and sustainable release abilities. Among vectors that have been constructed, lipid-based nanoparticles and polymer-based nanoparticles have represented enough encapsulation ability, good biocompatibility, and excellent circulation time. In addition, their complex structure consists of chemical modifications, nanoparticles, and scaffolds such as hydrogel and biological membranes that have more curative effects. Meanwhile, hydrogels have mesoporous structures that can encapsulate nanoparticles. The recombination of diverse delivery systems as described above seems to have a commendable clinical application prospect.

## Figures and Tables

**Figure 1 polymers-14-03338-f001:**
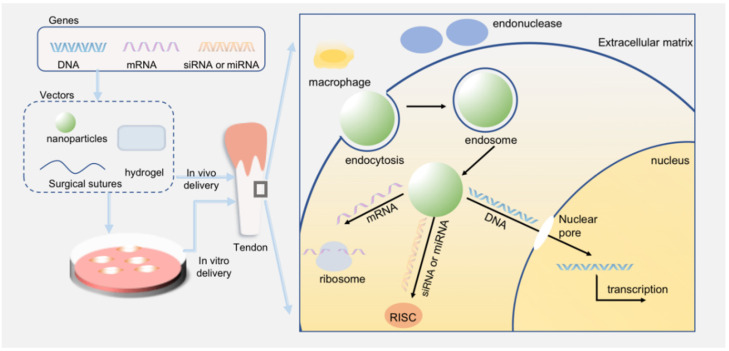
The process of gene therapy and barriers to the successful delivery of nucleic acids. Various genes can be delivered to tendons, including DNA, mRNA, siRNA, and miRNA. These drugs can be transfected with diverse delivery systems, such as nanoparticles, surgical sutures, and hydrogel. Gene therapy consists of two fundamental methods: in vivo and in vitro delivery. When they are released into the extracellular matrix, they have to overcome the degradation of endonucleases and the detection of immune systems. Then, they will be encapsulated in the cell by endocytosis, which induces the production of the endosome. Finally, mRNA will initiate translation; siRNA and miRNA must be loaded into the RNA-induced silencing complex (RISC), while DNA has to pass through the nuclear membrane and function in the nucleus.

**Figure 2 polymers-14-03338-f002:**
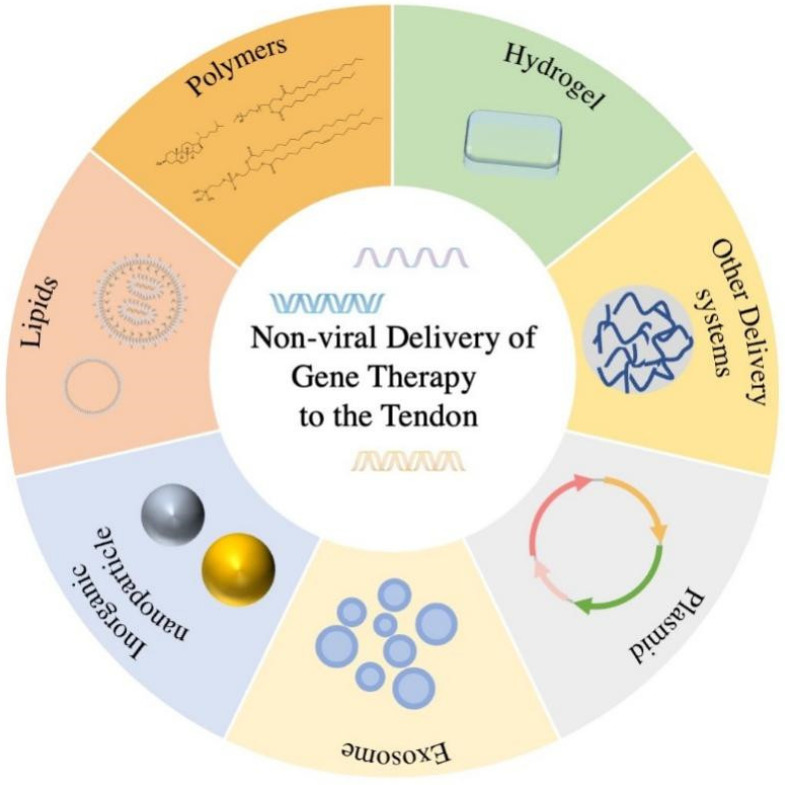
A graphical abstract figure of this review. The non-viral delivery of gene therapy to the tendon consists of plasmids, exosomes, inorganic nanoparticles, lipids, polymers, hydrogels, and other delivery systems.

**Figure 3 polymers-14-03338-f003:**
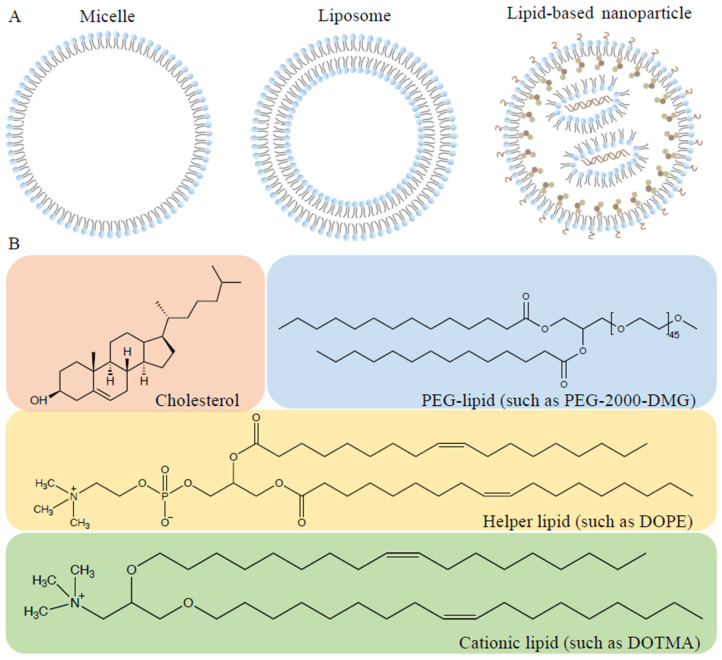
Structures of lipids and lipid-based nanoparticles. (**A**) Lipoplexes include the micelle, which is a single-layer structure, the liposome, which consists of two lipid layers, and the lipid-based nanoparticle. (**B**) The representative components of the Lipid-based nanoparticle. Four chemical structures constitute lipid-based nanoparticles, including cholesterol, PEG-lipids, helper lipids, and cationic or ionizable lipids that are particularly related to the efficiency of LNPs.

**Figure 4 polymers-14-03338-f004:**
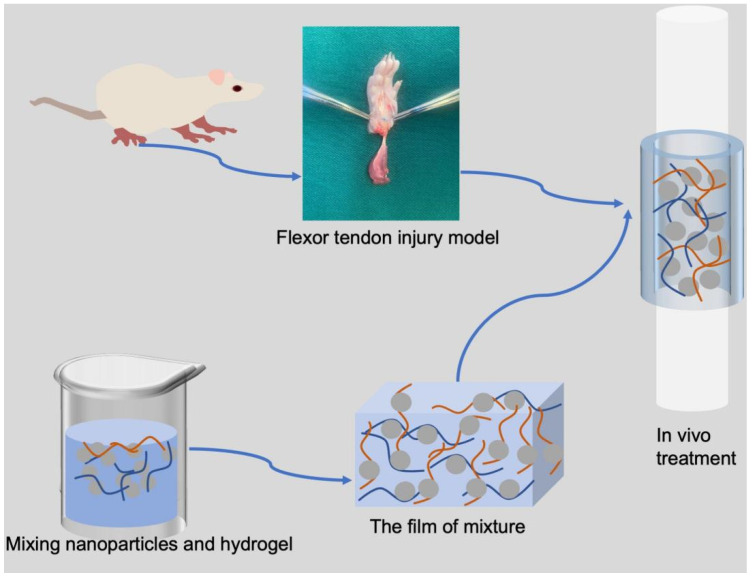
This schematic diagram represents the application of hydrogel in vivo. As a mesoporous polymer network that can encapsulate drugs, the hydrogel is usually encapsulated with diverse nanoparticles that have been mixed with target genes. After the mixture is frozen at room temperature, a film is formed. The flexor tendon injury model could be wrapped in hydrogel film.

**Table 1 polymers-14-03338-t001:** siRNAs have been used in tendon healing models.

Reference	Year	Target Gene	Function on Tendon	Type of Study	Title
[57]	2006	Runx2 Cbfa1	The silencing of Runx2/Cbfa1 inhibits the formation of heterotopic ossification.	in vitro	Adenovirus-mediated transfer of siRNA against Runx2/Cbfa1 inhibits the formation of heterotopic ossification in animal model
[58]	2007	cadherin-11	Cell–cell junctions and alignment of collagen fibrils are mediated by cadherin-11, which promotes tendon formation.	in vitro	Tendon development requires regulation of cell condensation and cell shape via cadherin-11-mediated cell-cell junctions
[59]	2008	COMP	COMP protects chondrocytes against apoptosis via elevating the proteins of the IAP family.	in vitro	Cartilage oligomeric matrix protein protects cells against death by elevating members of the IAP family of survival proteins
[60]	2008	IL-1beta	The silencing of IL-1beta regulates MMP-13, which is also affected by fatigue loading.	in vitro	Coordinate regulation of IL-1beta and MMP-13 in rat tendons following sub-rupture fatigue damage
[61]	2009	NCX	NCX involves in the role of fibroblasts during tendon healing	in vitro	Involvement of Na+/Ca2+ exchanger in migration and contraction of rat cultured tendon fibroblasts
[62]	2009	APC	APC promotes the proliferation of tenocytes and the synthesis of collagen1.	in vitro	Activated protein C mediates a healing phenotype in cultured tenocytes
[63]	2010	Rnux2 Smad4	The inhibition of Runx2 and Smad4 could prevent heterotopic ossification.	in vitro	Non-virus-mediated transfer of siRNAs against Runx2 and Smad4 inhibit heterotopic ossification in rats
[64]	2011	collagen V	Collagen V α1 plays an important role in tendon regeneration.	in vitro	Col V siRNA engineered tenocytes for tendon tissue engineering
[65]	2012	Wnt5a	The Wnt5a-RhoA pathway plays an important role in uniaxial mechanical tendon-induced osteogenic differentiation.	in vitro	Uniaxial mechanical tension promoted osteogenic differentiation of rat tendon-derived stem cells (rTDSCs) via the Wnt5a-RhoA pathway
[66]	2013	AMPKα1	HGF inhibits GF-β1-induced myofibroblastic differentiation via AMPK.	in vitro	Hepatocyte growth factor inhibits TGF-β1-induced myofibroblast differentiation in tendon fibroblasts: role of AMPK signaling pathway
[67]	2013	Mohawk	The inhibition of MKX would downregulate COL1A1 and TNXB and upregulate SOX9.	in vitro	Transcription factor Mohawk and the pathogenesis of human anterior cruciate ligament degradation
[68]	2013	ERK2	Tendon adhesion will be regulated by the inhibition of ERK2.	in vivo	Prevention of Tendon Adhesions by ERK2 Small Interfering RNAs
[69]	2015	TGIF1	TGIF1 could prevent tendon-to-bone from chondrogenic differentiation.	in vitro	TGIF1 Gene Silencing in Tendon-Derived Stem Cells Improves the Tendon-to-Bone Insertion Site Regeneration
[70]	2015	Pin1	Pin1 plays an important role in the progression of TSPCs aging.	in vitro	The role of Pin1 protein in aging of human tendon stem/progenitor cells
[71]	2015	scleraxis	Scleraxis is vital to the differentiation of TSCs to tenocytes.	in vitro	Dexamethasone inhibits the differentiation of rat tendon stem cells into tenocytes by targeting the scleraxis gene
[72]	2015	TGIF1	Rats perform better functions after being treated with TGIF1-siRNA BMSCs.	in vitro	Silencing of TGIF1 in bone mesenchymal stem cells applied to the post-operative rotator cuff improves both functional and histologic outcomes
[73]	2015	TNF-α	NF-κB, MMP1, MMP9, COX-1, and COX-2 which involve in inflammation may be downregulated.	in vivo	Targeted knockout of TNF-α by injection of lentivirus-mediated siRNA into the subacromial bursa for the treatment of subacromial bursitis in rats
[74]	2017	RelA/p65	p65 plays a core role in fibrosis by inhibiting cell proliferation and the expression of ECM.	in vitro	RelA/p65 inhibition prevents tendon adhesion by modulating inflammation, cell proliferation, and apoptosis
[75]	2017	ANGPTL4	ANGPTL4 serves as a multifunctional protein to regulate cell migration and proliferation.	in vitro	Angiopoietin-like 4 Enhances the Proliferation and Migration of Tendon Fibroblasts
[76]	2018	serpine1	The inhibition of serpine1 promotes the activity of MMP, which could protect tendons against adhesion.	in vitro	Serpine1 Knockdown Enhances MMP Activity after Flexor Tendon Injury in Mice: Implications for Adhesions Therapy
[77]	2018	scleraxis	Scx regulates several mechanosensitive proteins involved in adhesion.	in vitro	Novel roles for scleraxis in regulating adult tenocyte function
[78]	2018	scleraxis	Scx enhances the level of tenomodulin.	in vitro	Scleraxis is a transcriptional activator that regulates the expression of Tenomodulin, a marker of mature tenocytes and ligamentocytes
[79]	2018	FOXP1	FOXP1 promotes self-renewal of TSPCs by decreasing E2F1, pRb and cylin D1.	in vitro	Downregulation of FOXP1 correlates with tendon stem/progenitor cells aging
[80]	2020	Flightless I	Flii could reduce the proliferation and migration of human tenocyte.	in vitro	Increasing the level of cytoskeletal protein Flightless I reduces adhesion formation in a murine digital flexor tendon model
[81]	2020	Collagen III	polyDMAEA-siRNA polyexes show more promising efficiency compared to PEI-siRNA.	in vitro	Synthesis and Formulation of Four-Arm PolyDMAEA-siRNA Polyplex for Transient Downregulation of Collagen Type III Gene Expression in TGF-β1 Stimulated Tenocyte Culture
[82]	2021	CLK2 DYRK1A	SM04755 reduces inflammation and enhances tenocytes differentiation by inhibiting CLK2 and DYRK1A	in vitro	SM04755, a small-molecule inhibitor of the Wnt pathway, as a potential topical treatment for tendinopathy
[83]	2021	Smad3	the inhibition of transforming. Growth factor-β (TGF-β1)/Smad2/3 signal pathway could enhance tendon healing.	in vivo	Inhibition of Smad3 promotes the healing of rotator cuff injury in a rat model
[84]	2021	ITGA9	Tenascin-C promotes the regeneration of tendons via ITGA9-mediated migration of STSCs.	in vitro	Tenascin-C regulates migration of SOX10 tendon stem cells via integrin-α9 for promoting patellar tendon remodeling
[85]	2022	Smad3	The delivery system serves as an effective antiadhesion barrier, which could also decrease inflammation.	in vivo	Self-Healing Hydrogel Embodied with Macrophage-Regulation and Responsive-Gene-Silencing Properties for Synergistic Prevention of Peritendinous Adhesion
[86]	2022	IKKβ	Blocking KKβ/NF-κB pathway in vivo could treat RCT well.	in vivo	Inhibition of IKKβ/NF-κB signaling facilitates tendinopathy healing by rejuvenating inflamm-aging induced tendon-derived stem/progenitor cell senescence
[87]	2022	COX	The inhibition of COX could transform M1 to M2.	in vivo	Morphological changes of macrophages and their potential contribution to tendon healing

## Data Availability

Not applicable.

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
