# Peer review of "Non-Viral Delivery of Gene Therapy to the Tendon"

_polymers, 2022, doi:10.3390/polym14163338_

Round 1
Reviewer 1 Report
Dear authors,
Here are my comments for your review paper:
1. "In 2022, gene therapy, as a method based on the innate character of disease, rejuvenate.[15] [16] January 7, 2022, application of Zolgensma for clinical trials was approved, as announced by The Drug Evalution Center of the state Food and Drug Administration of China. [17] January 26, 2022, AAV-RPE65 also received acceptance. [18]". Try to reformulate as it is a little bit long and confusing.
2. Try to check different paragraphs where words are written with capital or small and it is not needed. Check also the English as some phrasea are too long and confusing.
3. Which are the novelty and added value with respect to literature?
4. I would add some more schemes or figures to support the applications for tendon injury.
5. You could also add a graphical abstract for the review.
6. More examples of hydrogels for tendon repair could be added: for i.e. Janus Tough Adhesive (JTA), a novel material. I think there is a mistake in Hydrogel chaper: Fabrin is fibrin, right?
7. "And the second group is a ternary polymer including DMAEMA, 2-propylacrylic acid (PAA), and butyl 433 methacrylate (BMA)..." Is this a terpolymer or a graft copolymer?
8. Which is the experience of the authors with the non-viral vectors topic for tendon? Or gene delivery?
Reviewer 2 Report
I would like to ask the authors a question regarding self-citation. If I understand correctly, references 6, 20, and 34 are the authors' previous opinions. These are not experimental articles, but rather expert opinions that have already been published in the literature. Am I correct in assuming that this is a re-citation of opinions already stated? Perhaps we should either express this explicitly as "we have previously asserted (concluded) that...", or we should not refer to these papers, but rather present these conclusions in a slightly more explicit form.
Overall, the review has many problems with the layout of the text. The main problems are related to the design of links, I have given only a few examples of this in the first comment.
Lines 53-54, 74, 89, 129, 132, 138. “of disease, rejuvenate.[15] [16]”, “research topics. [23]”, “healing properties. [34]”, “recognization receptor.[38]“, “cell membrane.[39]“, “of nuclear.[40]“. Multiple typos in the text when the dot is in the sentence, and references related to the statement are after the dot, for example, I listed PART of such typos.
Line 185. “30% of all.and”. Misprint.
Line 262. “toxicity.91”. Misprint.
Line 264. “interactions.92”. Misprint.
Lines 315-376. Problems with text formatting.
Line 327. “lipid nanoparticles(LNPs) are…”. The sentence begins with a lowercase letter.
Lines 528-532. Why are the names of polymers capitalized in this place?
Lines 896-898. References 140 and 141 refer to the same article.
Round 2
Reviewer 2 Report
I'm glad the authors did such a great job of improving the article. As presented, the article can be published.